# Exploring Ghanaian nurses knowledge and application of bio-ethical principles in postoperative pain management

**Moses Banoya Tia** [1]*, **Lydia Aziato**[2], **Gladys Dzansi** [2]

**1** Methodist University College Ghana, Accra, Ghana, **2** University of Ghana, Accra, Ghana

* mbtia@st.ug.edu.gh

**Data Availability Statement:** All relevant data are within the paper.

**Funding:** authors received no funding for this work.

## Abstract

Managing postoperative pain require good understanding of the bio-ethical principles in order to preserve patients' rights. Bio-ethical principles in health care include autonomy, beneficence, justice and nonmaleficence. It is important that health care professionals understand that patients in pain have the right to satisfactory management. Good insight on ethical principles and how they relate to pain management places the nurse on a better pedestal to manage postoperative pain effectively. However, there is scanty literature on the level of Ghanaian nurses' knowledge and application of bio-ethical principles in postoperative pain management. Therefore, the study objectives were to: explore nurses' understanding of the bio-ethical principles in postoperative pain management; explore how nurses apply bioethical principles in postoperative pain management. The study employed qualitative exploratory descriptive design. Purposive sampling technique was used to recruit participants from the surgical wards. Semi-structured interview guide was designed for data collection. Data saturation was reached at the fourteenth participant. Thematic analysis method was used and themes emerged inductively. Three main themes identified through inductive content analysis of data were: beneficence, autonomy and justice. Findings showed that nurses had some appreciable level of knowledge of the fundamental principles related to ethics and applied them in postoperative pain management. Nurses knew their duties in advocating for patients. Patients rights to refuse treatment was also appreciated by some nurses. Nurses also demonstrated humanity by helping patients financially to settle hospital debts which explicitly shows the empathetic characteristics of nurses. The study concluded that nurses are knowledgeable in bioethical principles underpinning post operative pain management and also applied these principles when caring for surgical patients.

## Introduction

Ethics is important in the nursing profession. Ethics guide nurses in their practice on a daily basis to distinguish between right and wrong when the correct path is unclear in professional care environment [1]. Nurses are expected to work in an ethical manner in order to meet the individual needs of patients and enhance satisfaction. Nurses who are knowledgeable about

**Competing interests:** The authors have declared that no competing interests exist,

ethical theories may be better prepared for ethical nursing practice as they will have thorough knowledge of the most appropriate ethical approaches in a given situation [2]. Ethics also enable nurses with the power to influence the day-to-day decisions that are made regarding patients care [3].

The underlying universal principles of ethics important for nursing practice are based on the obligation to do good (beneficence); do no harm (nonmaleficence); providing equal and fair treatment for all persons (justice); and defending individual determination (autonomy) [4]. These ethical principles provide a framework for the analysis and resolution of moral problems encountered in the delivery of healthcare. They form the basis for all codes of ethics and also the foundation of considerations for all professional groups in healthcare, including the nursing profession [5]. Thus, in making decision to care for patients with postoperative pain, these principles influence nurses to achieve the objectives of care.

Beneficence is an important ethical principle that underpins nursing care and is basically the duty to do good to patients while considering their desires [6]. Beneficence and non-maleficence are closely related. Beneficence is the principle of doing good [6–8]. Beneficence involve the active promotion of benevolent acts such as goodness, kindness, charity [9]. It may also include the injunction from inflicting harm. Nurses are obliged to implement actions that benefit patients and their support persons. Intentional harm is never acceptable in nursing. However, placing a patient at risk of harm has different facets. Unintentional harm is when the risk could not have been anticipated. Nonmaleficence is avoiding the causation of harm [6–8].

It was argued that providing comfort can be deemed as an essential component of beneficence [10]. Comfort needs of patients should be viewed as beneficence since it involves providing measures to relief pain. A phenomenological study conducted to explore nurses' and patients' views of comfort needs showed that comfort is an indispensable human need whether in illness or in good health [11]. Comfort was seen as a state of being free from suffering and being in a calm environment. This therefore presupposes that nurses' efforts to provide comfort for their patients can be viewed as benevolent acts and professional duties.

Beneficent actions of nurses could be thwarted when they are confronted with workload and less staffing since time may not be sufficient to perform the core duties of care. A study to evaluate the impact of understaffed nursing shifts on patient outcomes in Australia showed that patient care quality dwindled with associated effects such as surgical wound infections, urinary tract infections, deep vein thrombosis, pressure injuries, and pneumonia [12] suggesting that such patients' comfort would be compromised.

Autonomy refers to the right to make one's own decisions. In contemporary discourse it has broad meanings, including individual rights [13, 14], privacy, and choice [15]. Autonomy entails the ability to make a choice free from external constraints. Patient autonomy is viewed as giving chance to make choices from options that have been made known to them regarding their own treatment. Autonomy involves allowing the patients to decide on what health interventions they wish to be provided for them [16].

Nurses who adhere to the principle of autonomy understand that each patient is unique and has the right to behave differently and make choices and set goals concerning treatment. Honouring the principle of autonomy require that the nurse respect patients' decisions even if those decisions are not in the best interest of the patient [17]. The nurse has to also be considerate. Autonomy is violated in nursing care when a patient complain of pain is considered as exaggeration or demand for attention and thus, is not given the attention deserved. Also, when decision making processes are weak, patient autonomy is easily offended [18].

Pain is described as subjective phenomenon and is better perceived, assessed and treated just as the individual in pain explains it [19]. This explanation could be better understood in the context of patient autonomy. A study also elucidated that the subjective nature of pain is

best treated when nurses understand and view pain as the patients says it is [20]. By this, patients will feel comfortable since their views are accepted. This could depict that nurses acknowledge patients' autonomy regarding their expression of pain.

An important aspect of enhancing autonomy is through patient education. It was found in a study that patients empowerment is best achieved through education so that they are in a position to make informed choices [21]. The study further noted that patients became knowledgeable about their conditions and were better able to carefully make informed decisions with education. These findings buttress the role of empowerment in providing patients with the opportunity to have control of their health. By this, patients may feel they have roles in ensuring their own good health and thus, participate actively towards recovery. Other studies, however, showed that patients preferred shared decision making rather than being completely autonomous [22].

Justice is often referred to as fairness and equality [17]. Nurses must decide how much time they have to spend with each patient, taking patient needs into consideration, and then fairly distributing resources accordingly. Justice in nursing deals with fair treatment of patients and ensuring that the rights of individual patients are upheld [23]. Alzheimer Europe describes justice as the moral obligation to act on the basis of fair adjudication between competing claims [7]. Justice is further linked to entitlement, and equality. Justice is also seen as impartiality and objectivity towards patients, irrespective of their social status, race, or colour [24].

Advocacy can largely enhance justice. Advocacy roles played by nurses ensures safe practices in health care settings; advocacy for patients help in mitigating risks arising from sudden changes in patients' conditions and possible harm that may occur from misjudgement in the treatment of patients by other professionals [25]. Furthermore, protecting patients' rights is an aspect of advocating in the interest of patients [26].

A cross-sectional study in Uganda on nurses' knowledge of ethics showed that only 18 out of 114 participants (15.8%) of the nurses and midwives obtained a score ≥50% in the ethics knowledge assessment test. Nurses who had diploma or higher education were less likely to score below 50% in the knowledge test as compared with nurses who had obtained only certificates [27]. It was further noted that nurses with higher education scored higher in ethics knowledge test [27]. Studies have also shown that nurses with long years of service, higher educational attainment and belonging to a professional nursing association exhibited high professionalism attributes such as autonomy, accountability, advocacy, collaboration and collegiality [28].

There is an established positive correlation between professional autonomy and higher education on attitude towards caring for dying patients [29] which presupposes that higher carrier development and years of experience are most likely to positively influence professional ethical standards in nursing care including ethical decision making for postoperative pain management. Studies also revealed that nurse practitioners who had knowledge of professional ethics such as autonomy were confident in healthcare delivery decisions which improved their quality of work [30]. This presupposes that as professional nurses know ethics relating to pain management, they will give satisfactory care.

The authors extrapolated from previous studies on ethical principles since there is absolutely no single study on nurses' knowledge of bio-ethical principles regarding pain management. It is against this background that there is need to explore how nurses view the ethical principles and apply in nursing care of surgical pain.

## Methods

### Design

In order to explore nurses' knowledge of the ethical principles and their application to postoperative pain management, the researchers made use of qualitative exploratory descriptive

design. This is characterised by ability to provide comprehensive summaries of a phenomenon or of events in everyday language. This design was used in order to enable the researchers allow flexibility for participants' responses, and also owing to the fact that little is known about the topic under study [31]. This design allows for use of open-ended questions for exploring of a process, a variable or a concept that is not yet fully known or understood [31]. Exploratory descriptive design is amenable to health environments research because it provides factual responses to questions about how people feel, about a space, and what reasons they have for using features of a space [32]. The design therefore, enabled the researchers to gain understanding of the nurses' level of knowledge and how they applied bio-ethical principles in pain management by providing flexible questions where respondents could express themselves in detail.

## Setting

This study was conducted in a Regional hospital in Ghana. The facility serves as a major referral centre for surgical conditions from all other facilities under the region. It has a large surgery department which is divided into male and female wards. Patients who undergo various major and minor surgical interventions are received and managed in this department. Research participants were obtained from both male and female adult surgical wards of the hospital. This setting was chosen because it has a good number nursing staff who participate in nursing surgical patients.

## Population and sampling technique

Out of the total population of 176 nurses in the regional hospital at the time of data collection, 81 were registered general nurses and 95 were enrolled nurses. The target population was registered general nurses with diploma in nursing. The sampling technique used to recruit participants into the study was homogenous purposive sampling. It is a sampling technique in which the researcher aims at achieving homogeneity of participants. By this, the researcher seeks to obtain a sample that has similar characteristics such as same level of education and occupation. This sampling technique was suitable since it enabled the researchers with the opportunity to select respondents who were believed to have knowledge of postoperative pain management. This was because the researchers needed participants who could respond to the interview questions and were also willing to participate. Registered general nurses with a minimum of diploma certificates and one year minimum working experience were eligible to participate. Nurses on various forms of leave were excluded from the study. The researchers approached potential participants on duty and explained the purpose of the study to gain consent for interview. All the participants who were contacted reponded to the interviews willingly. Interviews were conducted until the thirteenth participant when data saturation was reached. One more participant was interviewed to ensure that saturation was actually reached. Therefore, a total of 14 participants were interviewed.

## Data collection and process

The researchers designed a semi-structured interview guide for data collection process. The literature review in related topics in ethics informed the formulation of the interview guide. Also, expert opinion from the second and third authors were sought in developing the interview guide. Individual interviews were conducted on participants and responses were audio taped. The interview guide comprised of open-ended questions which allowed participants to express their views well. Some of the main interview questions asked were: "explain some of the bioethical principles," "How do you make use of the bioethical principles when managing

patients' pain in the surgical ward?" Iterative questions were also posed in order to gain clarity on ambiguous response. In situations where there was need to gain depth of explanations, further probing were done on participants. Interviews were conducted in English language and transcribed verbatim. Also, field notes were taken in order to help confirm responses from participants. Fields notes included non-verbal cues and expressions. Data collection took place from January to February 2018. Each interview lasted for an average of 45 minutes. The first researcher carried out the data collection.

## Data analysis

Transcribed data were read through severally by the first researcher in order to gain full understanding of text. Content analysis method was used in analysing transcribed data. In content analysis, important themes and patterns that emerge from participants are identified [33]. It involves identifying and condensing meaning units of words or texts [34]. The approach was used to identify emerging themes from the data. Codes were generated from data and themes were subsequently obtained inductively. Verbatim quotes were used to support claims made by the researchers. Data transcription, coding, and analysis were done by the first author while the second and third authors verified results and ensured that the participants views were accurately represented. This was done by reviewing the analysed data carefully by comparing verbatim quotes with data from the participants.

## Rigor

Credibility establishes whether or not the findings of a research represent plausible information obtained from respondents' original views in the data obtained [35–37]. Member checking, the heart of credibility [38] was employed. Analysed and interpreted data were given to respective participants to evaluate the interpretations made by the researchers. Transferability is described as the degree to which qualitative research findings easily transferred to other contexts with other participants [35]. Transferability was ensured by selecting participants who were believed to be able to respond to the interview questions that would address the research objectives. Participants' privacy and anonymity were also guaranteed to enable them express their views without fear of being heard and identified by others. This approach enhanced honesty of the participants regarding information they were given in the interview process. These ensured dependability of participants' information. Also, data collection instrument dependability was guarranteed through a pre-testing of the interview guide. This was done in a different health facility that had similar characteristics as the facility the study was conducted. Responses of participants was used to reshape the interview guide to ensure its clarity.

## Demographics

There were fourteen participants including three males and eleven females. Seven each of the participants were from male and female surgical wards. Participants' ages ranged from 24 to 40 years. Six were 35 years and above, and the rest were between the ages of 24 and 30 years. Six (6) were married while eight were singles. Five participants had more than 10 years of work experience, and nine had between one and ten years work experience. Nine participants had diploma in nursing and the rest had bachelors degrees in nursing.

## Research ethical consideration

Ethical clearance was obtained from Noguchi Memorial Institute for Medical Research Institutional Review Board and Ghana Health Service Ethics Review Committee with clearance

numbers CPN 016/17-18 and GHS-ERC: 012/10/17 respectively. Each participant signed written informed consent form willingly after they understood every detailed explanations made regarding the study. The forms emphasised confidentiality, anonymity, voluntariness and participant's right to withdraw from study at any time without consequences. Anonymity of participants was ensured by the use of codes, while confidentiality was maintained through invididual interviews in privacy according to participants' discretion.

## Results

Three main themes emerged from the data analysis. These are beneficence, autonomy and justice. Two sub-themes were obtained from "beneficence": financial support for patients, and kindness with patients. The theme "autonomy" also had the following sub-themes: patients' empowerment, and nurses' empowerment. The third theme also had two sub-themes, namely, non-discrimination of patients and advocating for patients. Verbatim quotations are used to back the subthemes in the presentation of the results.

### Beneficence

#### Financial support for patients

Some of the participants said that they rendered various forms of help to patients. They said it was part of their inherent moral duty. Some examples of help that participants mentioned were giving money to patients to buy medications: "*Sometimes we the nurses buy food for patients who cannot afford.*" (**FN7**). Other nurses indicated that they supported patients with money for food. One of the nurses said that hunger is sometimes a reason for their pain and thus, helped those in serious need: *"some of the patients may afford the analgesics but are hungry. You can't also give some of these drugs on empty stomach. We just sometimes look for money to get them food." (*MN4**).

Some patients who were unable to pay their bills during discharge were helped by nurse: "...o*ther times we help in other things after discharge, like money for them to settle some bills.*" (**FN1**).

Nurses also extended help to patients who had surgical operations and did not have enough money for analgesics. One nurse narrated:

> "*for some of our patients, I sometimes add up my few coins to be able to get their analgesics from the pharmacy because you don't even see their relatives tell them that more money is needed for their patient's treatment.*" (**FN14**).

#### Kindness with patients

Participants also said that it is their duty to make patients comfortable by managing their pain. One participant said:"*. . .making the patient feel free of pain, that is to avert or alleviate the pain or reduce it as much as I can.*" (**FN7**).

Another participant described her duties to include actions that bring about good results and recovery of patients. An example mentioned was administering pain medication before dressing surgical wounds:

> "*it is simply ensuring that your nursing actions are intended to produce good results so that the patient can recover fast. . .I will give pain killer maybe thirty minutes before I dress the wound.*" (**FN13**)

Others stated that they go ahead and administer pain medications whenever the patients complain of pain. Nurses said they would not withhold analgesics from patients in pain:

"*I am always kind to the patients with their analgesics once I hear of any complain of pain from my patients.*" (**MN4**)

"*I have no reason to keep analgesics in the name of addiction when my patients are showing signs of pain. I give them PRN with the slightest complain to avoid worsening pain*" (**MN 9**).

Participants also said that they tried to be generous with patients' prescribed medications:

"*We try to ensure that we are generous with medications that are prescribed. . .without discriminating.*" (**FN7**).

Some of the participants also indicated that spending more time outside of their schedules with patients especially in cases of staff shortage. A participants said: "*sometimes we just stay longer for the sake of the patients, just to give extra helping hand when we are having shortage. Sometimes for a whole extra shift outside my schedule.*" (**FN13**)

### Autonomy

**Patients' empowerment.** Participants said they were always giving patients opportunities in decision making: "*. . .the patient has the opportunity to make suggestions about himself regarding their pain treatment.*" (**MN4**)

Some respondents also noted that making treatment options available was integral empowerment strategies for patients: "*The patient is supposed to be aware of all the things that may happen to him so that he can choose whatever he wishes.*" (**MN9**)

Other participants said that educating patients on their conditions and the available treatment would make them get involved in their treatment: "*. . .you should explain to the patient what he should know so that he can be involved in whatever you are doing to him.*" (**MN4**).

Seven of participants considered that the subjective nature of pain was better managed by allowing patient make informed decisions. One of them narrated:

"*The patient has the right to express his pain and request for medication because he alone can tell how severe the pain is. For me I think it is important we even allow or tell them to inform us when they are in pain. They are even helping us by telling us how they feel*" (**FN14**)

Only one participant noted the need to document patient's refusal of procedures in their attempts to exercise autonomy:

"*. . .if you are going to do something for the patient, if the patient does not want it, he doesn't want it. You cannot force the patient to take it. Yours is that you have to document it.*" (**MN9**).

Other participants mentioned the need to make senior staff aware of patient's wilful refusal of certain treatment:

"*Like a patient can say I don't like this medication. You can't force him to take it. . . then you may let your in-charge know that you did your best but patient refused*" (**FN1**)

**Nurses' empowerment.**  Some of the nurses said that they were able to make decisions on patients' care when doctors were not on the wards:

"*When the doctors are not around sometimes you just know that if I do this it will work so sometimes you just do it. You just take your own independent decisions and it help the patients*" (**FN7**)

Other participants were able to assess patients with acute pain using the numeric rating scale and made prescriptions of analgesics for patients:

"*Sometimes we are able to assess and score patients pain, especially those with acute pain and write medication. We go to the pharmacy and explain and by seeing the pain score they serve us and we give to the patient and document.*" (**FN14**)

FN7 said that they were involved in decision making. She noted that making decisions with doctors concerning patients treatment was considered teamwork and the doctors recognised nurses' ability in decision making:"*Sometimes on rounds we make inputs and decisions with the doctors. They appreciate nursing is autonomous body. . .*" (**FN7**)

## Justice

**Non-discrimination of patients.**  A participant said that non-discriminatory attitude towards patients is achieved by showing respect and treating them equally:

". . .*I consider all patients as equal and so I respect and treat them without discrimination. They are given their medications as they deserve especially the pain medications*" (**FN11**)

Some participants indicated that attending to patients should not be on the basis of patients' social class or race:

". . .*we treat them equally despite their colour, race, whatever. . .so far as you are working with the individual and as it says, you should treat all of them equally, so you will work on that*" (**FN13**)

Some also said that since pain is subjective, patients should be allowed to express their feelings about pain and not disciminate how they may want to put it. One of them stated:". . .*and the patient has the right to express their pain. We know that pain is subjective and patients should be allowed to say how they feel.*"(**FN2**).

A participant also said they do not withhold prescribed pain medications from patients since it could be seen as discrimination:

". . .*If the pain killers are there, you have to administer. . .otherwise, when others complain and you give, it will be said that you discriminate*" (**MN9**).

Many participants said there should be fair treatment of patients, such as being attentive to patients' needs and being non-discriminatory:

"*You should make sure you treat them equally. Don't say because this one's father is a minister should get that, and this one should get that. . . I think being fair sometimes could be giving them the kind of attention they need*" (**FN3**).

Others said nurses should not be selective when attending to patients:

"*If you are not attending to them don't attend to them at all. If you are doing stuff, do it but if you attend to one call, if the next person calls, you are supposed to go or else they will start saying something.*" (**FN10**).

**Advocating for patients.**   Participants noted that nurses are involved in advocating for patients in situations where they may not have the ability to do so. One of them narrated:

"*I remember we had a patient who came here with road traffic accident and for just some few hours of stay in the hospital, they billed him about seven hundred cedis (GHC 700) which he had nothing to pay. It was only paracetamol that we gave and the paracetamol too he bought so we said no to this injustice. We had to send the folder back ourselves and explained to them and the bill was corrected to forty-seven cedis (GHC 47.00) and he paid and left.*" (**FN2**).

Some nurses alluded that they explained to patients during the preoperative period about their rights to request for pain medications: "*. . .and we also try to explain to the patients that they can make request for analgesics when they come back from theatre*" **FN13.**

Some of the younger staff also said that they had to convince their senior nurses on duty before they administered analgesics to patients who were considered to exagerate their pain. A nurse stated:

"*our seniors who have worked for long sometimes tell us that this patient is faking his pain. But sometimes I take the pain to check the patient and persuade my senior on duty to allow me give at least paracetamol*" **Fn1.**

## Discussion

Nurses were able to explain beneficence, autonomy, nonmaleficence, and justice in various ways. Nurses demonstrated how these principles relate to pain management. All the nurses had a minimum qualification of diploma in general nursing, possibly depicting the role of higher education on nurses understanding of nursing ethics. Previous studies conducted proved that nurses with higher degrees in nursing had good understanding of ethical principles [27]. Contrary findings were revealed in Ghana as they found that health professionals including nurses had low knowledge of bio-ethical principles [39]. Similarly, an Ethiopian study found that nurses had inadequate knowledge of patient autonomy [28]. It could be deduced that incorporating ethical principles in surgical nursing curriculum could enhance the provision of ethically acceptable nursing care.

Nurses helped patients who could not afford food and medications with money. They said that such acts were benevolent to patients which is supported by previous findings which noted that beneficence encompasses acts of charity and mercy [9]. The inability of some patients to afford certain things while on admission identified in the study could be a reflection of the poor socioeconomic status of the region [40] which makes access to quality healthcare difficult. Even though the National Health Insurance Scheme (NHIS) is available, some medications and treatment are not covered and thus, patients have to pay for such treatment.

The nurses also made efforts to ensure patients' comfort as part of their nursing roles. Nurses spent extra work time for their patients' wellbeing. Other studies have explained that

beneficence is doing good and sacrificing for patients [7, 9]. Policies for compensating nurses who provide extra support for patients may motivate other staff. It was also found that nurses sacrificed their energy and time to care for overwhelming high numbers of patients at the expense of their own health. They indicated that it was a form of doing good to patients. Nurses being overburdened with work could lead to a state of moral distress once they may be unable to accomplish all their roles despite their knowledge of expected duties as indicated in related studies [41].

Nurses in this current study said that they provided treatment options to patients and allowed them to make decisions. This finding is consistent with the assertion that autonomy is enabling patients to make decisions on health care interventions they want to receive [16]. Furthermore, nurses in the current study said that the subjective nature of pain was best treated when autonomy of patients was upheld which supports an earlier finding that pain experience is subjective and nurses need to give patients individualised attention [20]. If patients' autonomy is viewed as a right, then it can be deduced that nurses will respect patients' views. This is because it will place obligation on nurses to appreciate patients' autonomous decisions. Quality nursing care will therefore, be enhanced [42] since patients will feel valued.

The study also established that nurses empower patients through education on their disease conditions and making treatment options available. This finding supports an earlier study which showed that patients' empowerment began when professionals recognised that patients were in control of their conditions and were able to make informed decisions after educating them [21]. This may imply that patient education not only provides knowledge, but also empower patients to take part in their care which can reduce morbidity as it was found that patients with hypertension had improved healthy lifestyle behaviours and decreased blood pressure after they were educated [43].

The nurses also said that they were able to make decisions independently in the absence of prescribers. It also identified that nurse autonomy include making independent judgement and having freedom to function [13]. The nurses in this study knew their decision making roles. Earlier studies indicated that nurses were aware of their decision making roles [44]. Some nurses made decisions with doctors and collaborated in patients' pain management. This can be presumed that nurses' autonomy is being enhanced in patient care. It also imply that teamwork has the potential of increasing nurses' autonomy which is consistent with a previous study [42]. Providing measures that enable nurses to participate in decision making and collaboration in the healthcare environment will enhance autonomy and help improve quality patient care since nurses' confidence will increase [30].

Participants explained that they were being non-discriminatory and meted fair treatment to patients. Nurses further said that justice is being impartial irrespective of the patient's colour, race or social status [45]. It was also found that being impartial to patients could be achieved through paying attention to their needs, and respecting their rights to treatment [46]. As nurses are impartial and acknowledge patients' rights to pain treatment, nurses become generous with prescribed analgesics.

Nurses also explained that advocating for patients and defending vulnerable ones were also aspects of ensuring fair treatment of patients. Nurses who advocate for patients safeguard interests and well-being of patients [25]. Previous study also found that protecting patients' rights was part of advocacy [26]. A recent study also concluded that, advocacy roles played by nurses may enhance patient autonomy [47], suggesting that patients who are advocated for may end up having knowledge of their rights to treatment and therefore, will expect fair treatment from nurses.

### Limitations of the study

The study made use of only surgical wards of a single health facility which could limit findings from being representative of Ghanaian nurses' views. Only registered general nurses participated in the study purposively. This has limited other category of nurses who could have different and divergent views of the ethical principles. This has therefore, limited generalizability of the study findings.

### Implications of the study

Considerable knowledge of bio-ethical principles by nurses has the tendency to impart holistic of postoperative management which has been a long standing problem. This must be considered since conventional approaches to pain management do not typically yield desired outcomes for patients. The moral obligation to attend to patients in pain should therefore, be intrinsic and cultivated by professional nurses through internalisation of the bio-ethical principles in health through training. In-service training programmes in bio-ethics can be a good option.

## Conclusion

General nurses in the study understood that nurses and patients' autonomy has the potential of enhancing better patients' pain management outcome. Nurses also appreciated the need to be fair to patients by providing them with the required analgesics based on the ethical principles of justice and beneficence. Nurses went the extra mile to assist patients in various ways to secure medications and settle hospital treatment bills due to their compassionate nature. Hospital authorities therefore, need to continue to encourage nursing staff and provide them with some sort of incentives so that nurses feel movitated and acknowledged.

## Acknowledgments

Authors express sincere thanks to the management and staff of the hospital in which the study was carried. Their willingness to accommodate us for the study is highly appreciated.

## Author Contributions

**Conceptualization:** Moses Banoya Tia, Lydia Aziato.

**Data curation:** Moses Banoya Tia.

**Formal analysis:** Moses Banoya Tia.

**Writing – original draft:** Moses Banoya Tia.

**Writing – review & editing:** Moses Banoya Tia, Lydia Aziato, Gladys Dzansi.

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
