## [Decision Letter · Decision Letter 0]

21 Mar 2022

PONE-D-21-23036

Exploring Ghanaian Nurses knowledge and application of bio-ethical principles in postoperative pain management.

PLOS ONE

Dear Dr. Tia,

Thank you for submitting your manuscript to PLOS ONE. After careful consideration, we feel that it has merit but does not fully meet PLOS ONE’s publication criteria as it currently stands. Therefore, we invite you to submit a revised version of the manuscript that addresses the points raised during the review process.

A 'Response to Reviewers' letter that responds to each point raised by the academic editor and reviewer(s). You should upload this letter as a separate file labeled 'Response to Reviewers'.A marked-up copy of your manuscript that highlights changes made to the original version. You should upload this as a separate file labeled 'Revised Manuscript with Track Changes'.An unmarked version of your revised paper without tracked changes. You should upload this as a separate file labeled 'Manuscript'.

We look forward to receiving your revised manuscript.

Kind regards,

Prof. Ritesh G. Menezes, M.B.B.S., M.D., Diplomate N.B.

Academic Editor

PLOS ONE

Journal Requirements:

2. Please include a copy of the interview guide used in the study, in both the original language and English, as Supporting Information, or include a citation if it has been published previously.

Reviewers' comments:

Reviewer's Responses to Questions

**Comments to the Author**

1. Is the manuscript technically sound, and do the data support the conclusions?

Reviewer #1: Partly

Reviewer #2: Yes

Reviewer #3: Yes

Reviewer #4: Yes

Reviewer #5: No

Reviewer #6: Partly

2. Has the statistical analysis been performed appropriately and rigorously? 

Reviewer #1: No

Reviewer #2: N/A

Reviewer #3: N/A

Reviewer #4: Yes

Reviewer #5: N/A

Reviewer #6: No

3. Have the authors made all data underlying the findings in their manuscript fully available?

Reviewer #1: Yes

Reviewer #2: Yes

Reviewer #3: Yes

Reviewer #4: Yes

Reviewer #5: No

Reviewer #6: Yes

4. Is the manuscript presented in an intelligible fashion and written in standard English?

Reviewer #1: Yes

Reviewer #2: Yes

Reviewer #3: Yes

Reviewer #4: Yes

Reviewer #5: No

Reviewer #6: No

5. Review Comments to the Author

Reviewer #1: Tia et al. conducted a study on, “Exploring Ghanaian Nurses knowledge and application of bio-ethical principles in postoperative pain management”, in which they aim to explore nurses’ understanding and application of the bio-ethical principles in postoperative pain management. They show that nurses are knowledgeable in bioethical principles underpinning post-operative pain management and also applied these principles when caring for surgical patients. In my opinion the study can be improved by incorporating the following points:

1. On page 5, the author has mentioned regarding statements regarding justice. In my opinion, this doesn’t support the rationale of the topic. The introduction is a bit lengthy due to author mentioning details that are not closely related to the objectives/rationale, hence, revise the introduction and remove such statements.

2. The objectives of the study are not clearly mentioned. It should be stated as what the authors aim to find and investigate from this study. A mention of primary and secondary objectives will also be appreciated.

3. Mentioning that just the researchers made a semi-structured questionnaire is insufficient. More statements are required regarding did any senior author or expert opinion was considered for the questionnaire?

4. Data regarding the number of nurses or beds can be included to improve the quality of study setting portion of methodology.

5. A major issue, associated to point#4, is that the sample size is very small. It is difficult to assess whether the results are good enough as we have no data regarding the number of nurses or beds.

6. Was any software, website or any particular mathematical equation used in sample size calculation?

7. The authors have mentioned regarding a pre-test performed on the nurses. Where have they mentioned the results or any other details regarding this test? What was the impact of the test of the final questionnaire? There are ambiguities in the questionnaire drafting process which needs to be clearly highlighted by the author.

8. The statistical analysis part lacks a mention of how the data was analyzed. The statistical analysis section needs to be completely rewritten. What software’s were used? What percentages, mean, mode, or ranges and in what units were calculated through the statistical packages used?

9. What is the mean age? What is the mean work experience?

10. The authors need to clearly mention how many participants did they approach and how many responded.

11. A figure or table summarizing demographics of the participants would enhance the quality of the study.

12. In the first paragraph of discussion, more statements are required to explain and support the findings regarding bioethics knowledge that the nurses’ have.

13. There is bias when the authors use purposive sampling. This limits the data and the results cannot be extrapolated as results cannot be generalized. Why hasn’t this been mentioned in the limitations of the study?

14. It is important to compare and contrast the findings with previous studies with a similar aim and rationale. One such study is, “https://www.ncbi.nlm.nih.gov/pmc/articles/PMC7429762/”

Reviewer #2: Thank you for the opportunity to review this paper regarding

Ghanaian Nurses knowledge and application of bio-ethical principles in postoperative pain management. I was able to get a good sense of the nurses’ role in Ghana, the challenges such as burden of increased workload faced by them. I was surprised to learn that the nurses’ spent from their pockets to feed the needy and also bought pain medications for them. The kind gesture is beyond the ethical principles described in books.

I would like to have more clarification on the following points.

• pain is described as subjective phenomenon and is better perceived, assessed and treated just as the individual in pain explains it’ . Please correct it as ‘Pain is …’

• ‘total of 14 participants were interviewed’ can be corrected as ‘Total of …’

• “All the nurses had a minimum qualification of diploma in general nursing, possibly depicting the role of higher education on knowledge of nursing ethics.” Do you mean to say diploma nurse did not have enough knowledge? Please explain how the knowledge of diploma nurses are measured? The verbatim statements reflect that the nurses were practising ethically. If knowledge is not separately measured, the term ‘knowledge’ can be removed from title. Knowledge can be measured as a continuous variable. Since the authors have not measured knowledge using a questionnaire, it is better to remove the term from the title to avoid confusion to the reader.

Reviewer #3: The topic of bio-ethical principles is fundamental to nurses, particularly as nurses are continuously considered one of the most trusted professionals. Understanding what nurses know about these principles and how they apply them to their practice can help provide insight and guidance for future scholarship. The paper is clearly outlined and easy to follow for organization.

Existing literature supports the rationale for this exploratory study. The literature provides both foundational information as well as identifies gaps in the literature. This study aimed to lessen that gap.

The study is methodologically sound based on the aim of the study. The qualitative review process of transcription, coding, and analysis are all sound. The identified themes are supported by adequate illustrations.

I would suggest the manuscript is suitable for publication. I would suggest further development of implications of the study. Perhaps consider where do we go from here? What's next? What is necessary to facilitate nurse knowledge and application of bio-ethical behavior.

Reviewer #4: Manuscript is well written, however there are few errors noted which needs to be rectified. In abstract there is mention about four themes which is supposed to be three. The sentence in the abstract....' Nurses also demonstrated humanity by helping patients financially to settle hospital debts which explicitly shows the empathetic characteristics of nurses.' is not convincing about the practice of bio-ethical principles in pain management.

Reviewer #5: Summary of the research and overall impression:

This is a qualitative study that explores Ghanaian nurses’ understanding and application of bioethical principles in postoperative pain management. There is a clear gap in the literature and a need for an exploration around this topic.

Major Issues:

The final sentence of abstract (Thus, nurses need motivation as they support patients in various ways, such as financial

assistance). This seems like an odd placement for this sentence and it is very unclear as to how it relates to knowledge of fundamental principles related to bioethics.

The authors present a framework for bioethical principles (nonmaleficience, beneficence, autonomy, and justice) early on in the paper in the introduction but the themes presented are not well applied to this framework.

Methods-Design: What are the key principles of a qualitative exploratory design? The authors list the benefits of this design but don't identify for the reader the main components of qualitative exploratory design.

Methods-Data analysis:The authors state that the data analysis was primarily completed by the first author and that the second and third authors verified results and ensured that the participants views were accurately represented. How did they the second and third authors verify results and ensure that views were accurately represented? This is unclear.

Results- Some of the themes identified and presented do not clearly support the purpose of the paper nor do they align with the framework presented in the introduction of the paper. It is very unclear how financial support for patients (subtheme of Generosity) provides evidence about nurses’ understanding and application of bioethical principles in postoperative pain management.

The subtheme patient’s empowerment underneath empowerment to enhance autonomy is probably the most well-aligned theme. However the sub theme of nurses’ empowerment is a confusing application of the results to the framework presented in the introduction.

It might be more helpful to develop and organize the themes under the four bioethical principles that are presented in the introduction and referred to throughout the paper.

The conclusion of the paper ends with an idea that nurses should receive incentives to feel motivated and acknowledged. It is unclear how that is related to nurses applying bioethical principles to pain control.

The manuscript needs significant editing for grammar, language and writing quality.

Minor Issues:

Methods- population and sampling technique: The authors should identify that they utilized a convenience sample.

The authors state in the abstract that there are four themes but only list three.

Reviewer #6: Suggest Bioethics is defined and referred to throughout the submission to link to nurses pain management

Page 5 spelling error. There are areas requiring greater clarity of expression (Page 5). The data analysis process is not transparent. The selection of the setting was not justified strongly. It was not apparent that semi structured interviews were used in any depth in the design section. What was the value in asking participants if they were married or single. Could have debated the act and ethics of nursing staff giving patients money. Could have presented key themes and inclusion and exclusion criteria in tables so these were more explicit. Page 14. What kind of decisions were made in the absence of prescribers and the relevance of this? Could have substantiated any generalisations for example these nurse had ethical probity whereas the literature states others do not?? Please improve conclusion to include to make this more substantial.

6. PLOS authors have the option to publish the peer review history of their article (what does this mean?). If published, this will include your full peer review and any attached files.

Reviewer #1: No

Reviewer #2: No

Reviewer #3: **Yes: **Tracy R. Vitale, DNP, RNC-OB, C-EFM, NE-BC

Reviewer #4: No

Reviewer #5: No

Reviewer #6: No

---

## [Author Response · Author response to Decision Letter 0]

4 Jul 2022

reviewer #1

1. On page 5, the author has mentioned regarding statements regarding justice. In my opinion, this doesn’t support the rationale of the topic. The introduction is a bit lengthy due to author mentioning details that are not closely related to the objectives/rationale, hence, revise the introduction and remove such statements. This has been addressed

2. The objectives of the study are not clearly mentioned. It should be stated as what the authors aim to find and investigate from this study. A mention of primary and secondary objectives will also be appreciated. The objective of the study is stated under the abstract of the paper.

3. Mentioning that just the researchers made a semi-structured questionnaire is insufficient. More statements are required regarding did any senior author or expert opinion was considered for the questionnaire? The second and third authors’ expert views and opinions were sought about the interview guide.

5. A major issue, associated to point#4, is that the sample size is very small. It is difficult to assess whether the results are good enough as we have no data regarding the number of nurses or beds

6. Was any software, website or any particular mathematical equation used in sample size calculation? The study was qualitative and so the sample size was not exactly predetermined with any statistical formulae. Sample size was arrived as interviews were conducted and at a point no new information was forthcoming from participants.

8. The statistical analysis part lacks a mention of how the data was analysed. The statistical analysis section needs to be completely rewritten. What software’s were used? What percentages, mean, mode, or ranges and in what units were calculated through the statistical packages used?

9. What is the mean age? What is the mean work experience? Content analysis was done and thus, no statistical software was required.

Content analysis does not require computing percentages, mean, mode or ranges and units

10. The authors need to clearly mention how many participants did they approach and how many responded. All the participants approached responded to the interview at will.

13. There is bias when the authors use purposive sampling. This limits the data and the results cannot be extrapolated as results cannot be generalized. Why hasn’t this been mentioned in the limitations of the study? This important limitation has been added 

Reviewer #2

• pain is described as subjective phenomenon and is better perceived, assessed and treated just as the individual in pain explains it’ . Please correct it as ‘Pain is …’ That has been addressed in the manuscript.

‘total of 14 participants were interviewed’ can be corrected as ‘Total of …’

It has been corrected to three themes in the abstract and in the results sections.

All the nurses had a minimum qualification of diploma in general nursing, possibly depicting the role of higher education on knowledge of nursing ethics.” Do you mean to say diploma nurse did not have enough knowledge? The verbatim statements reflect that the nurses were practising ethically. If knowledge is not separately measured, the term ‘knowledge’ can be removed from title. Knowledge cannot be measured in this study since it is qualitative. The title has therefore been slightly altered.

Reviewer #3

I would suggest further development of implications of the study. Perhaps consider where do we go from here? What's next? What is necessary to facilitate nurse knowledge and application of bio-ethical behavior. This has been addressed in the implications section. 

Reviewer #5

The authors present a framework for bioethical principles (nonmaleficience, beneficence, autonomy, and justice) early on in the paper in the introduction but the themes presented are not well applied to this framework. The themes emerged from the content analysis of data.

Methods-Design: What are the key principles of a qualitative exploratory design? The authors list the benefits of this design but don't identify for the reader the main components of qualitative exploratory design. It is addressed in page 6 (design section)

Methods-Data analysis: The authors state that the data analysis was primarily completed by the first author and that the second and third authors verified results and ensured that the participants views were accurately represented. How did they the second and third authors verify results and ensure that views were accurately represented? This is unclear. This is addressed in the analysis section-page 7

Results- Some of the themes identified and presented do not clearly support the purpose of the paper nor do they align with the framework presented in the introduction of the paper. It is very unclear how financial support for patients (subtheme of Generosity) provides evidence about nurses’ understanding and application of bioethical principles in postoperative pain management. Please Financial support is act of beneficence and is supported by previous study as in the discussion section.

The subtheme patient’s empowerment underneath empowerment to enhance autonomy is probably the well-aligned theme. However the sub theme of nurses’ empowerment is a confusing application of the results to the framework presented in the introduction. That is addressed.

It might be more helpful to develop and organize the themes under the four bioethical principles that are presented in the introduction and referred to throughout the paper. That is addressed in the results section of the paper.

The conclusion of the paper ends with an idea that nurses should receive incentives to feel motivated and acknowledged. It is unclear how that is related to nurses applying bioethical principles to pain control. That part is taken out of the work please.

The manuscript needs significant editing for grammar, language and writing quality. Proof reading has been done to correct these mistakes.

Methods- population and sampling technique: The authors should identify that they utilized a convenience sample. Purposive sampling was rather used. It is indicated.

The authors state in the abstract that there are four themes but only list three. That is rectified in the abstract and the results.

REVIEWER #6

Page 5 spelling error. There are areas requiring greater clarity of expression (Page 5). Grammatical expressions and errors have been addressed.

The data analysis process is not transparent. Quotes from participants were used to bark claims made by authors and this seem to be in line with content analysis. 

The selection of the setting was not justified strongly. Justification of setting has been stated.

---

## [Decision Letter · Decision Letter 1]

8 Sep 2022

PONE-D-21-23036R1Exploring Ghanaian Nurses knowledge and application of bio-ethical principles in postoperative pain management.PLOS ONE

Dear Dr. Tia,

Thank you for submitting your manuscript to PLOS ONE. After careful consideration, we feel that it has merit but does not fully meet PLOS ONE’s publication criteria as it currently stands. Therefore, we invite you to submit a revised version of the manuscript that addresses the points raised during the review process.

Please submit your revised manuscript by Oct 23 2022 11:59PM. Please include the following items when submitting your revised manuscript:A 'Response to Reviewers' letter that responds to each point raised by the academic editor and reviewer(s). You should upload this letter as a separate file labeled 'Response to Reviewers'.A marked-up copy of your manuscript that highlights changes made to the original version. You should upload this as a separate file labeled 'Revised Manuscript with Track Changes'.An unmarked version of your revised paper without tracked changes. You should upload this as a separate file labeled 'Manuscript'.If applicable, we recommend that you deposit your laboratory protocols in protocols.io to enhance the reproducibility of your results. Protocols.io assigns your protocol its own identifier (DOI) so that it can be cited independently in the future. For instructions see: https://journals.plos.org/plosone/s/submission-guidelines#loc-laboratory-protocols. Additionally, PLOS ONE offers an option for publishing peer-reviewed Lab Protocol articles, which describe protocols hosted on protocols.io. Read more information on sharing protocols at https://plos.org/protocols?utm_medium=editorial-email&utm_source=authorletters&utm_campaign=protocols.

We look forward to receiving your revised manuscript.

Kind regards,

Prof. Ritesh G. Menezes, M.B.B.S., M.D., Diplomate N.B.

Academic Editor

PLOS ONE

Journal Requirements:

Please review your reference list to ensure that it is complete and correct. If you have cited papers that have been retracted, please include the rationale for doing so in the manuscript text or remove these references and replace them with relevant current references. Any changes to the reference list should be mentioned in the rebuttal letter that accompanies your revised manuscript. If you need to cite a retracted article, indicate the article’s retracted status in the References list and also include a citation and full reference for the retraction notice.

Reviewers' comments:

Reviewer's Responses to Questions

**Comments to the Author**

1. If the authors have adequately addressed your comments raised in a previous round of review and you feel that this manuscript is now acceptable for publication, you may indicate that here to bypass the “Comments to the Author” section, enter your conflict of interest statement in the “Confidential to Editor” section, and submit your "Accept" recommendation.

Reviewer #3: All comments have been addressed

Reviewer #4: All comments have been addressed

Reviewer #6: All comments have been addressed

2. Is the manuscript technically sound, and do the data support the conclusions?

Reviewer #3: Yes

Reviewer #4: Yes

Reviewer #6: Partly

3. Has the statistical analysis been performed appropriately and rigorously? 

Reviewer #3: N/A

Reviewer #4: N/A

Reviewer #6: Yes

4. Have the authors made all data underlying the findings in their manuscript fully available?

Reviewer #3: Yes

Reviewer #4: Yes

Reviewer #6: Yes

5. Is the manuscript presented in an intelligible fashion and written in standard English?

Reviewer #3: Yes

Reviewer #4: Yes

Reviewer #6: Yes

6. Review Comments to the Author

Reviewer #3: 1. What are the aims of objectives of this study. They should be clearly delineated rather than suggested.

2. What percentage of the nursing staff was included? How was the sample size determined? Was it until saturation was achieved?

3.

Reviewer #4: The author has now related the ethical principles to pain management which is convincing the reader. Following grammatical errors need to be corrected.

Demographics: Page 8 - there is no need to mention the number in the bracket as it is the repetition of the word.

‘There were fourteen (14) participants including three (3) males and eleven (11) females’

Spelling error - Participants

Page 10: There is a repetition of word- give

Reviewer #6: This has improved and the authors have responded to my individual comments. I do not think they have responded to the other reviewers comments in full for example; there are still no clear aims and objectives apparent in the introduction or the methods section, therefore these cannot be revisited to establish if there were achieved. It would also have been helpful for me to be better signposted within the manuscript as to where all changes had been made so that comparisons could be made between the original and the resubmitted manuscripts.

7. PLOS authors have the option to publish the peer review history of their article (what does this mean?). If published, this will include your full peer review and any attached files.

Reviewer #3: No

Reviewer #4: No

Reviewer #6: No

---

## [Author Response · Author response to Decision Letter 1]

15 Sep 2022

Reviewer # 3

1. What are the aims of objectives of this study? They should be clearly delineated rather than suggested. response:The aim/objective of the study is stated in the abstract.

2. What percentage of the nursing staff was included? This has been addressed by stating the total number of nursing staff. On page 7

3. How was the sample size determined? Was it until saturation was achieved?This has been addressed on page 7 as “interviews were conducted until the thirteenth participant when data saturation was reached. One more participant was interviewed to ensure that saturation was actually reached. Therefore, a total of 14 participants were interviewed.”

Reviewer # 4

1. Page 8 - there is no need to mention the number in the bracket as it is the repetition of the word. The numbers in bracket has been removed

2. There were fourteen (14) participants including three (3) males and eleven (11) females’ Spelling error - Participants. The spelling error has been addressed. 

3. page 10: There is a repetition of word- give. The repeated word "give"has been deleted

---

## [Editor Report · Decision Letter 2]

28 Sep 2022

PONE-D-21-23036R2

Exploring Ghanaian Nurses knowledge and application of bio-ethical principles in postoperative pain management.

PLOS ONE

Dear Dr. Tia,

Thank you for submitting your manuscript to PLOS ONE. After careful consideration, we feel that it has merit but does not fully meet PLOS ONE’s publication criteria as it currently stands. Therefore, we invite you to submit a revised version of the manuscript that addresses the points raised during the review process.

Please submit your revised manuscript by 7-October-2022. Additional time will not be provided and if the deadline is crossed, then a resubmission will be recommended. Please include the following items when submitting your revised manuscript:

A 'Response to Reviewers' letter that responds to each point raised by the academic editor and reviewer(s). You should upload this letter as a separate file labeled 'Response to Reviewers'.

A marked-up copy of your manuscript that highlights changes made to the original version. You should upload this as a separate file labeled 'Revised Manuscript with Track Changes'.An unmarked version of your revised paper without tracked changes. You should upload this as a separate file labeled 'Manuscript'.

We look forward to receiving your revised manuscript by 7-October-2022.

Kind regards,

Prof. Ritesh G. Menezes, M.B.B.S., M.D., Diplomate N.B.

Academic Editor

PLOS ONE

Journal Requirements:

Please review your reference list to ensure that it is complete and correct. If you have cited papers that have been retracted, please include the rationale for doing so in the manuscript text or remove these references and replace them with relevant current references. Any changes to the reference list should be mentioned in the rebuttal letter that accompanies your revised manuscript. If you need to cite a retracted article, indicate the article’s retracted status in the References list and also include a citation and full reference for the retraction notice.

Additional Editor Comments:

A. Make sure that the following points are addressed adequately and clearly.

1) defined objectives or research questions; 2) description of the sampling strategy/sample size determination, including rationale for the recruitment method, participant inclusion/exclusion criteria and the number of participants recruited; 3) a discussion of potential sources of bias; and 4) a discussion of limitations

B. Minor revisions

1) presentation style corrections (one example: methods-design-6th line:- "as suggested by (34)"; another example: "According to (39–41)")

---

## [Author Response · Author response to Decision Letter 2]

4 Oct 2022

1) defined objectives or research questions.The objectives of the research is stated in the abstract as “Therefore, the study objectives were to: explore nurses’ understanding of the bio-ethical principles in postoperative pain management; and explore how nurses apply bioethical principles in postoperative pain management.” 

2) description of the sampling strategy/sample size determination, including rationale for the recruitment method, participant inclusion/exclusion criteria and the number of participants recruited response: Homogenous sampling technique has been described as “It is a sampling technique in which the researcher aims at achieving homogeneity of participants. By this, the researcher seeks to obtain a sample that has similar characteristics such as same level of education and occupation.” sample size determination is stated “Interviews were conducted until the thirteenth participant when data saturation was reached. One more participant was interviewed to ensure that saturation was actually reached.” 

Inclusion and exclusion criteria is stated: “Registered general nurses with a minimum of diploma certificates and one year minimum working experience were eligible to participate. Nurses on various forms of leave were excluded from the study.”

Number of participants is mentioned as “Therefore, a total of 14 participants were interviewed.”

4) a discussion of potential sources of bias and limitations. response: These are indicated under the “limitations of the study” of the manuscript.

presentation style corrections (one example: methods-design-6th line:- "as suggested by (34)"; another example: "According to (39–41)"). response: These mistakes are addressed, example, line 6 under “design” is changed to … responses, and also owing to the fact that little is known about the topic under study (34).

Comment on references. response: references have been updated and errors corrected.

---

## [Editor Report · Decision Letter 3]

7 Oct 2022

Exploring Ghanaian Nurses knowledge and application of bio-ethical principles in postoperative pain management.

PONE-D-21-23036R3

Dear Dr. Tia,

We’re pleased to inform you that your manuscript has been judged scientifically suitable for publication and will be formally accepted for publication once it meets all outstanding technical requirements.

Kind regards,

Prof. Ritesh G. Menezes, M.B.B.S., M.D., Diplomate N.B.

Academic Editor

PLOS ONE

---

## [Editor Report · Acceptance letter]

11 Oct 2022

PONE-D-21-23036R3 

Exploring Ghanaian Nurses knowledge and application of bio-ethical principles in postoperative pain management. 

Dear Dr. Tia:

I'm pleased to inform you that your manuscript has been deemed suitable for publication in PLOS ONE. Congratulations! Your manuscript is now with our production department. 

Kind regards, 

on behalf of

Prof. Dr. Ritesh G. Menezes 

Academic Editor

PLOS ONE